# FewVS: A Vision-Semantics Integration Framework for Few-Shot Image Classification

## ABSTRACT

Some recent methods address few-shot image classification by extracting semantic information from class names and devising mechanisms for aligning vision and semantics to integrate information from both modalities. However, class names provide only limited information, which is insufficient to capture the visual details in images. As a result, such vision-semantics alignment is inherently biased, leading to suboptimal integration outcomes. In this paper, we avoid such biased vision-semantics alignment by introducing CLIP, a natural bridge between vision and semantics, and enforcing unbiased vision-vision alignment as a proxy task. Specifically, we align features encoded from the few-shot encoder and CLIP's vision encoder on the same image. This alignment is accomplished through a linear projection layer, with a training objective formulated using optimal transport-based assignment prediction. Thanks to the inherent alignment between CLIP's vision and text encoders, the few-shot encoder is indirectly aligned to CLIP's text encoder, which serves as the foundation for better vision-semantics integration. In addition, to further improve vision-semantics integration at the testing stage, we mine potential fine-grained semantic attributes of class names from large language models. Correspondingly, an online optimization module is designed to adaptively integrate the semantic attributes and visual information extracted from images. Extensive results on four datasets demonstrate that our method outperforms state-of-the-art methods. *The code is available in supplementary materials.*

## CCS CONCEPTS

• **Computing methodologies → Object recognition**.

## KEYWORDS

Few-shot image classification; modality alignment; optimal transport

## 1 INTRODUCTION

Few-shot image classification (FSIC) aims to recognize novel query samples by leveraging limited support samples. The prevalent paradigm for addressing FSIC is to learn a representative visual prototype, which is close to query samples of the same class and far away from query samples of other classes. However, the scarcity of support samples results in biased visual prototypes inevitably. To

tackle this problem, existing methods [8, 26, 44, 45, 48, 50] propose to extract semantic features from class names via a pretrained text encoder [13, 35, 36] as prior information of novel classes, and then design different cross-modal interaction modules to integrate the visual and semantic features for improving the FSIC performance. However, these methods ignore the inherent modality gap between the visual and semantic modalities, thus only achieving modest integration performance.

Recognizing this limitation, recent methods [8, 26, 45, 50] introduce various modality alignment mechanisms to bridge this modality gap for improved integration. Additionally, these methods leverage the semantic branch of CLIP, a vision-language pretrained model, as a powerful text encoder to obtain reliable semantic features. For instance, BMI [26] explicitly aligns the few-shot encoder with CLIP's text encoder by minimizing the discrepancies in latent distributions between the visual and semantic modalities, as well as the cross-modal reconstruction errors. SemFew [50] designs a semantic revolution module to augment class names to encode robust semantic features, achieving more accurate alignment between vision and semantics. However, the semantic information derived from class names is limited, which falls short of capturing the visual details of diverse images. Consequently, this leads to biased alignment between vision and semantics, producing suboptimal integration outcomes.

Actually, CLIP [36] can serve as a natural bridge between vision and semantics. It is trained on a vast dataset comprising over 400 million image-caption pairs using a contrastive learning paradigm, ensuring an inherent alignment between its vision and text encoders. In this paper, leveraging CLIP's inherent alignment, we enforce vision-vision alignment as a proxy task to avoid biased vision-semantics alignment. To be specific, we align features encoded from the few-shot encoder and CLIP's vision encoder on the same image at the training stage, by learning a linear projection layer with an optimal transport-based training objective. Due to the inherently aligned nature of CLIP's vision and text encoders, the few-shot encoder is indirectly aligned with CLIP's text encoder. In this way, we can perform CLIP-like vision-semantics integration using the few-shot encoder and CLIP's text encoder. Notably, compared with the heavy CLIP's vision encoder, the few-shot encoder is designed to be lightweight and can be trained from scratch using methodologies tailored for FSIC, such as meta-learning [16], to specifically learn and extract task-specific visual information.

To further improve the vision-semantics integration at the testing stage, we mine fine-grained semantic attributes for each class name from large language models, such as GPT-3 [4]. Such semantic attributes can describe potential visual details and provide richer prior information for novel classes than class names. However, the process of mining these semantic attributes is agnostic to the visual modality, possibly introducing noisy semantic attributes that mismatch novel classes. Therefore, drawing inspiration from [19], we

design an online optimization module to adaptively integrate these fine-grained semantic attributes with the visual information from query images. This is achieved through online optimizing a learnable importance weight for each semantic attribute's classification contribution using the labels of support samples as supervision. Through this process, we can identify the positive/negative impact of each semantic attribute on our classification objective and then re-weight its contribution, which guides an adaptive integration for more effective query classification.

The main contributions of this paper are summarized as follows:

- We propose a novel **Few**-shot **V**ision-**S**emantics integration framework, namely FewVS, to integrate vision and semantics for improving FSIC. In FewVS, we indirectly achieve unbiased vision-semantics alignment by introducing CLIP and enforcing vision-vision alignment as a proxy task, with an optimal-transport based training objective.
- To further enhance integration performance, at the testing stage, we mine fine-grained semantic attributes for each class by querying large language models, and design an online optimization module to adaptively integrate vision and semantics.
- Extensive experiments on four public benchmarks (*mini*ImageNet, *tiered*ImageNet, CIFAR-FS, and FC100) demonstrate that our FewVS outperforms state-of-the-art methods.

## 2 RELATED WORK

Existing FSIC methods can be mainly divided into two categories, *i.e.*, vision-based methods and semantics-based methods.

**Vision-based methods** leverage support samples to extract class-related features for classification. To achieve this, several methods [3, 16, 38, 51] with different optimization strategies learn an initial model that can rapidly adapt to novel tasks with a few steps of updating. However, due to limited support samples, such methods often suffer from overfitting. To address this issue, some methods [9, 31, 39, 47, 53] learn a metric space where novel samples are classified utilizing a prevalent prototype-based classifier [39]. Other methods [19, 27] focus on acquiring a robust few-shot encoder by employing a transformer-based backbone [15] and incorporating techniques such as distillation [17] and self-supervised learning [6, 18].

**Semantics-based methods** [8, 26, 44, 45, 48–50] extract semantic information from class names and integrate visual and semantic information to improve FSIC performance. Recent methods [8, 26, 45, 50] introduce implicit or explicit modality alignment mechanisms to bridge the gap between visual and semantic modalities for improved integration performance. For example, Xu *et al.* [44] introduced a conditional variational autoencoder (CVAE) to generate visual features conditioned on semantic features, which implicitly aligns the visual and semantic modalities in the latent space. Li and Wang [26] enforced explicit alignment by aligning latent distributions of the visual and semantic modalities and minimizing the cross-modal reconstruction errors. Zhang *et al.* [50] designed a semantic revolution module to augment class names to encode robust semantic features, achieving more accurate vision-semantics alignment. However, information derived from class names is insufficient to represent the visual details in diverse images. Consequently, the vision-semantics alignment in the above methods is inherently biased, leading to suboptimal integration outcomes.

## 3 PROPOSED METHOD

In this section, we first revisit the definition of FSIC and briefly introduce the testing pipeline of CLIP. We then elaborate on the training (Section 3.2) and testing (Section 3.3) pipelines of FewVS.

### 3.1 Preliminary

*3.1.1* **Problem definition**. In a typical FSIC scenario, we consider a base set $\mathcal{D}_b = \{(I, y), y \in C_b\}$ and a novel set $\mathcal{D}_n = \{(I, y), y \in C_n\}$. Here, $I$ denotes an image sample and $y$ represents the label of $I$. $C_b$ and $C_n$ correspond to the base and novel class sets, respectively, with the condition that they are disjoint, i.e., $C_b \cap C_n = \emptyset$. To provide semantic information, we map $y$ to its class name $c$, such as 'House finch' and 'Jellyfish'.

Adhering to a conventional $N$-way $K$-shot setting [16, 39], each episode samples $N$ classes from $C_n$, with $K$ samples per class, to construct the support set: $\mathcal{S} = \{(I_i^s, y_i^s)\}_{i=0}^{N \times K}$. Similarly, $N \times Q$ samples from these $N$ classes are sampled to form the query set: $Q = \{(I_i^q, y_i^q)\}_{i=0}^{N \times Q}$. Notably, $\mathcal{S} \cap Q = \emptyset$. The aim of FSIC is to train a model on $\mathcal{D}_b$ with strong generalization performance on $Q$ by utilizing the labeled samples from $\mathcal{S}$.

*3.1.2* **Testing pipeline of CLIP**. CLIP demonstrates remarkable performance in zero-shot image classification which is a task similar to FSIC. At the testing stage, given a query image $I^q$ and a set of possible class names $\{c_i\}_{i=1}^N$, CLIP firstly encodes visual and semantic features from the image and class names via its vision encoder $E_v(\cdot)$ and text encoder $E_s(\cdot)$, respectively. It then computes the cosine similarity scores between $I^q$ and each $c_i$ to integrate vision and semantics, relying on its inherent vision-semantics alignment property. Finally, the probability that $I^q$ belongs to class $i$ is computed via a softmax function:

$$P(y = i | I^q) = \frac{\exp(< E_v(I^q), E_s(c_i) >)}{\sum_{j=1}^N \exp(< E_v(I^q), E_s(c_j) >)} \quad (1)$$

where $< \cdot >$ is the cosine similarity function. Note that CLIP's two encoders share the same feature dimension. Due to the alignment nature between the vision and text encoders, CLIP can classify images of novel classes based only on semantic information about these novel classes.

### 3.2 Optimal Transport-based Alignment at the Training Stage

The overall training pipeline of FewVS is shown in Figure 1. In FewVS, we enforce vision-vision alignment as a proxy task to avoid biased vision-semantics alignment. This alignment equips FewVS with the capability to perform CLIP-like vision-semantics integration (as stated in Section 3.1.2) between a lightweight few-shot encoder and CLIP's text encoder. Specifically, we add a linear projection layer following the few-shot encoder to transform the features of the few-shot encoder into those of CLIP's vision encoder. To ensure the alignment quality, inspired by [5, 7, 33, 43], we use an

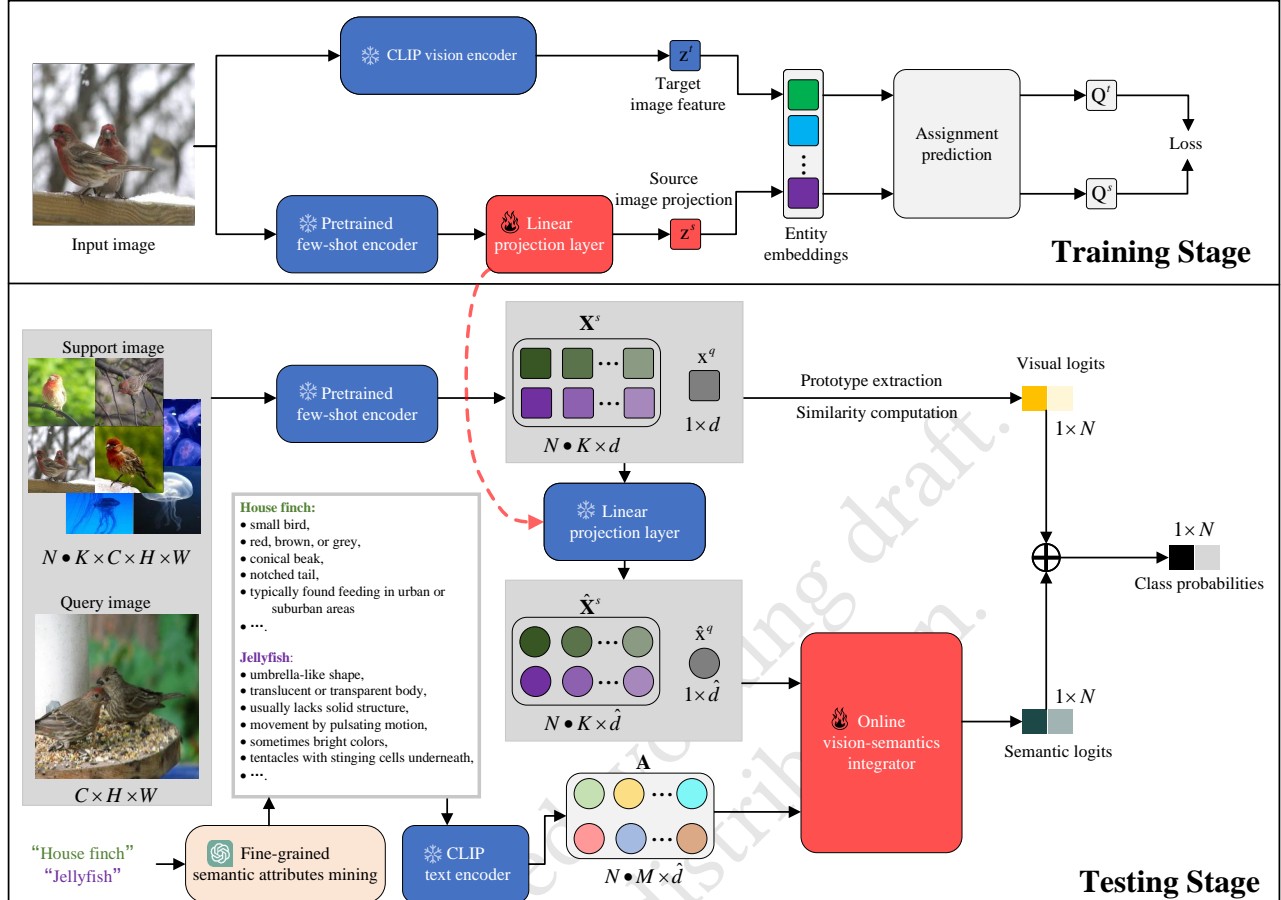

**Figure 1: Training and testing pipelines of FewVS. In the testing stage, we demonstrate the testing pipeline of an $N$-way $K$-shot episode. For better understanding, we illustrate the scenario with $N = 2$.**

optimal transport strategy to compute the assignment predictions of the features of the few-shot encoder and CLIP's vision encoder, respectively, thus obtaining two kinds of assignment predictions. Then, we train the linear projection layer by maximizing the consistency between these two kinds of predictions. Next, we further provide the details of the training pipeline.

*3.2.1 **Feature extraction.*** Given a batch containing $B$ images from $\mathcal{D}_b$, we feed them into frozen CLIP's vision encoder $E_v(\cdot)$, and extract the global pooled hidden features, denoted as $\mathbf{Z}^t = \{\mathbf{z}_1, ..., \mathbf{z}_B\} \in \mathbb{R}^{B \times d^t}$, which serves as the alignment target. Similarly, we feed the same images into the pretrained and frozen few-shot encoder $E_{few}(\cdot)$. Subsequently, we transform the resulting features $\mathbf{X} = \{\mathbf{x}_1, ..., \mathbf{x}_B\} \in \mathbb{R}^{B \times d^s}$ into the same dimension of $\mathbf{Z}^t$ via a learnable linear projection layer $h(\cdot)$, yielding the image projection denoted as $\mathbf{Z}^s$, which serves as the source of alignment:

$$\mathbf{Z}^s = \{\mathbf{z}_n^s \in \mathbb{R}^{d^t} | \mathbf{z}_n = h(\mathbf{x}_n), n = 1, ..., B\} \quad (2)$$

*3.2.2 **Assignment prediction.*** Different from previous works [2, 5, 33], we sample $L$ entities describing visual objects from a vast corpus (namely WordNet [30]), and encode their names via CLIP's

text encoder to construct an entity embedding set $\mathbf{E} = \{\mathbf{e}_1, ..., \mathbf{e}_L\} \in \mathbb{R}^{L \times d^t}$ which can be regarded as cluster centers. Then, we map $\mathbf{Z}^t$ and $\mathbf{Z}^s$ to $\mathbf{E}$ to obtain the assignments $\mathbf{Q}^t = \{\mathbf{q}_1^t, ..., \mathbf{q}_B^t\} \in \mathbb{R}^{B \times L}$ and $\mathbf{Q}^s = \{\mathbf{q}_1^s, ..., \mathbf{q}_B^s\} \in \mathbb{R}^{B \times L}$, respectively. Similar to [2, 5, 33], we optimize $\mathbf{Q}^t$ and $\mathbf{Q}^s$ to maximize the similarity between the features of CLIP's vision encoder and the entity embeddings and the similarity between the features of the few-shot encoder and the entity embeddings, respectively. The above process can be modeled as an optimal transport problem with the following objectives:

$$\max_{\mathbf{Q}^t \in Q} \mathrm{Tr}(\mathbf{Q}^{t\top}\mathbf{E}^\top\mathbf{Z}^t) + \epsilon H(\mathbf{Q}^t) \quad (3)$$

$$\max_{\mathbf{Q}^s \in Q} \mathrm{Tr}(\mathbf{Q}^{s\top}\mathbf{E}^\top\mathbf{Z}^s) + \epsilon H(\mathbf{Q}^s) \quad (4)$$

where $H$ is the entropy function, $H(\mathbf{Q}) = -\sum_{ij} \mathbf{Q}_{ij} \log \mathbf{Q}_{ij}$, and $\epsilon$ is a smoothness parameter of the assignment. Following [5], we implement an equal partition by constraining the matrix $Q$ to belong to the transportation polytope:

$$Q = \{\mathbf{Q} \in \mathbb{R}_+^{L \times B} | \mathbf{Q}\mathbf{1}_B = \frac{1}{L}\mathbf{1}_L, \mathbf{Q}^\top\mathbf{1}_L = \frac{1}{B}\mathbf{1}_B\} \quad (5)$$

where $\mathbf{1}_L$ denotes the vector of ones in dimension $L$. The two constraints in Eq. (5) ensure that on average each entity embedding is selected at least $\frac{B}{L}$ times in a batch.

The optimized assignments $\mathbf{Q}^{s*}$ and $\mathbf{Q}^{t*}$, namely the solutions of Prob. (3) and Prob. (4) over $Q$, take the form of a normalized exponential matrix [11]:

$$\mathbf{Q}^{s*} = \text{Diag}(\mathbf{u})\exp(\frac{\mathbf{E}^\top \mathbf{Z}^s}{\epsilon})\text{Diag}(\mathbf{v}) \tag{6}$$

$$\mathbf{Q}^{t*} = \text{Diag}(\mathbf{u})\exp(\frac{\mathbf{E}^\top \mathbf{Z}^t}{\epsilon})\text{Diag}(\mathbf{v}) \tag{7}$$

where $\mathbf{u}$ and $\mathbf{v}$ are renormalization vectors in $\mathbb{R}^L$ and $\mathbb{R}^B$, respectively, which are computed using the iterative Sinkhorn-Knopp algorithm [11]. More details are provided in the supplementary material.

*3.2.3 **Loss function***. To train the linear projection layer, following [5], we construct a "swapped" problem to predict the assignments of the features of CLIP's vision encoder based on the features of the few-shot encoder, and vice versa. Specifically, we predict $\mathbf{Q}^s$ from $\mathbf{Z}^t$ and $\mathbf{Q}^t$ from $\mathbf{Z}^s$, which can be formulated as the cross-entropy loss between the assignments and the probabilities that the corresponding features belong to the entity embeddings. The loss function is defined as:

$$\mathcal{L}_{align} = -\frac{1}{B}\sum_{b=1}^{B}\sum_{l=1}^{L}[\mathbf{Q}_{bl}^{s*}\log\mathbf{p}_{bl}^t + \mathbf{Q}_{bl}^{t*}\log\mathbf{p}_{bl}^s],$$
$$\text{where} \quad \mathbf{p}_{bl}^m = \frac{\exp(\mathbf{z}_b^m\mathbf{e}_l)}{\sum_{l=1}^{L}\exp(\mathbf{z}_b^m\mathbf{e}_l)}, \quad m \in \{s, t\} \tag{8}$$

This loss function ensures that the transformed features of the few-shot encoder contain the similar information with the features derived from CLIP's vision encoder, making the few-shot encoder indirectly aligned to CLIP's text encoder.

## 3.3 Vision-Semantics Integration at the Testing Stage

In this section, we first briefly introduce the overall testing pipeline of FewVS which is shown in Figure 1. Subsequently, we elaborate on the fine-grained semantic attribute mining and the online vision-semantics integrator.

*3.3.1 **Overall testing pipeline***. In a standard $N$-way $K$-shot episode with a support set $\mathcal{S} = \{(I_i^s, y_i^s)\}_{i=0}^{N\times K}$ and a query image $I^q$, we first feed both $\mathcal{S}$ and $I^q$ into the pretrained few-shot encoder $E_{few}(\cdot)$ to obtain visual support features $\mathbf{X}^s = \{\mathbf{x}_1^s, ..., \mathbf{x}_{N\cdot K}^s\} \in \mathbb{R}^{N\cdot K\times d}$ and visual query feature $\mathbf{x}^q \in \mathbb{R}^d$. Then, we transform $\mathbf{X}^s$ and $\mathbf{x}^q$ into $\hat{\mathbf{X}}^s \in \mathbb{R}^{N\cdot K\times\hat{d}}$ and $\hat{\mathbf{x}}^q \in \mathbb{R}^{N\cdot K\times\hat{d}}$ via a linear projection layer $h(\cdot)$, where $\hat{d}$ is the feature dimension of CLIP's text encoder $E_s(\cdot)$. Subsequently, we map the support labels back into class names, and mine $M$ fine-grained semantic attributes for each class using GPT-3, the details of which are provided later in Section 3.3.2. Then, we encode these semantic attributes via $E_s(\cdot)$ and obtain a semantic feature set $\mathbf{A} = \{\mathbf{a}_1, ..., \mathbf{a}_{N\cdot M}\} \in \mathbb{R}^{N\cdot M\times\hat{d}}$.

The final probability $P$ of a query sample $I^q$ belonging to class $n$ is computed from the weighted sum of visual logits $l^{vis} \in \mathbb{R}^N$ and

**American robin:**
- medium-sized bird
- orange or reddish breast
- gray  back and wings
- white lower belly
- often seen on the ground

**House finch:**
- small bird
- red, brown, or grey
- conical beak
- notched tail
- typically found feeding in urban or suburban areas

**Saluki:**
- slim and graceful sighthound
- long, thin limbs
- long, feathered ears
- short, silky coat
- varying colors

**Ladybug:**
- round or oval body shape
- red or orange with black spots
- black head with white patches
- six legs
- small size

**Figure 2: Example of the fine-grained semantic attributes mined from GPT-3.**

semantic logits $l^{sem} \in \mathbb{R}^N$:

$$P(y = n|I^q) = \frac{\exp(l_n^{vis} + \alpha l_n^{sem})}{\sum_{m=1}^{N}\exp(l_m^{vis} + \alpha l_m^{sem})} \tag{9}$$

where $\alpha$ is a weight factor determining the balance between the visual and semantic information.

To obtain $l_n^{vis}$, we follow [39] to extract the prototype of class $n$, denoted as $p(n)$, and compute the similarity between $\mathbf{x}^q$ and $p(n)$:

$$p(n) = \frac{1}{|\mathbf{X}_n^s|}\sum_{\mathbf{x}^s \in \mathbf{X}_n^s}\mathbf{x}^s, \quad l_n^{vis} = <\mathbf{x}^q, p(n)> \tag{10}$$

where $\mathbf{X}_n^s$ is a set of support features belonging to class $n$ in $\mathbf{X}^s$, and $< \cdot >$ denotes the cosine similarity function.

To obtain $l_n^{sem}$, a straightforward method is to compute the cross-modal similarity like CLIP:

$$\hat{p}(n) = \frac{1}{|\mathbf{A}_n|}\sum_{\mathbf{a} \in \mathbf{A}_n}\mathbf{a}, \quad l_n^{sem} = <\hat{\mathbf{x}}^q, \hat{p}(n)> \tag{11}$$

where $\mathbf{A}_n$ is a set of semantic features belonging to class $n$ in $\mathbf{A}$, and $\hat{p}(n)$ can be considered as the semantic prototype of class $n$. However, naively computing $l_n^{sem}$ is suboptimal, due to the noisy semantic attributes introduced by semantic attributes mining (detailed later in Section 3.3.2). Therefore, we design an online optimization module to adaptively integrate visual and semantic information for the computation of $l_n^{sem}$, the details of which are provided later in Section 3.3.3.

*3.3.2 **Fine-grained semantic attribute mining***. To further improve vision-semantics integration performance, we propose to mine fine-grained semantic attributes corresponding to each class name, which describe potential visual details of this class. Similar to [29], we automatically mine these fine-grained semantic attributes by prompting a large language model, such as GPT-3 [4], to query the visual details. We prompt the language model with the input: "`Q: What are the visual details for distinguishing a [class name] in a photo? A: There are several useful visual details of a [class name] in a photo:`", Further implementation details can be found in the supplementary material.

As depicted in Figure 2, the fine-grained semantic attributes describe potential visual details of a class, such as colors, shapes, object components, and scene contexts. Nevertheless, the process of

**Figure 3: Structure of the online vision-semantics integrator.**

mining these semantic attributes is agnostic to the visual modality and may introduce noisy semantic attributes that mismatch novel classes. For instance, we prompt GPT-3 with three class names "harvestman", "Newfoundland dog", and "bell", obtaining the corresponding semantic attributes. Among them, semantic attributes "lacks venom or silk glands" associated with "harvestman", "males typically weighing 130-150 pounds" associated with "Newfoundland dog," and "emits sound when moved by wind" associated with "bell" contain irrelevant noises. Obviously, semantic features encoded from these noisy attributes may have a negative impact on vision-semantics integration.

*3.3.3* **Online vision-semantics integrator**. As shown in Figure 3, the key point behind this module is to online learn an importance weight vector $\omega \in \mathbb{R}^{N \cdot M}$ to adjust the contribution of each semantic attribute in $\mathbf{A}$, with the support set $\mathcal{S}$ as supervision.

For each episode, we randomly initialize $\omega^0 = \mathbf{0} \in \mathbb{R}^{1 \times N \cdot M}$, and infer $\omega^*$ using labeled support samples via inner-loop optimization. To be more specific, we first construct a cross-modal similarity map by computing the pair-wise similarity between the transformed support features $\hat{\mathbf{X}}^s$ and semantic features $\mathbf{A}$ as $\mathbf{S}^s \in \mathbb{R}^{N \cdot K \times N \cdot M}$, where each element in $\mathbf{S}^s$ is obtained by $\mathbf{s}_{ij} = < \hat{\mathbf{x}}_i^s, \mathbf{a}_j >$, $i = 1, ..., N \cdot K$, and $j = 1, ..., N \cdot M$. Similarly, we compute the pair-wise similarity between the query image $I^q$ and $\mathbf{A}$ as $\mathbf{S}^q \in \mathbb{R}^{N \cdot M}$.

Then, we adjust the contribution of each semantic attribute to classification by performing broadcast and addition operations on $\omega$, thus obtaining an adjusted similarity map $\tilde{\mathbf{S}}^s = \mathbf{S}^s + [\mathbf{1}^{N \cdot K \times 1} \cdot \omega]$. In this way, the similarities associated with each semantic attribute are re-weighted via $\omega$. Subsequently, the semantic logits (denoted as $l_n^s$) of the $i$th support sample $I_i^s$ belonging to class $n$ can be computed as follows:

$$l_n^s = \frac{1}{|\tilde{\mathbf{S}}_{in}^s|} \sum_{\mathbf{s} \in \tilde{\mathbf{S}}_{in}^s} \mathbf{s} \qquad (12)$$

where $\tilde{\mathbf{S}}_{in}^s$ is a set of similarity logits in $\tilde{\mathbf{S}}^s$ associated with both $I_i^s$ and semantic attributes of class $n$. Then, the class prediction $y_i'$ of $I_i^s$ can be computed as: $y_i' = \text{softmax}(\{l_n^s\}_{n=1}^N)$.

Given that $\mathbf{y}' = \{y_i'\}_{i=1}^{N \cdot K}$ is dependent on $\omega$, we optimize $\omega$ to re-weight semantic attributes' classification contributions using support labels $\mathbf{y} = \{y_i\}_{i=1}^{N \cdot K}$ as supervision, and formulate an online optimization objective as follows:

$$\min_{\omega} \mathcal{L}_{CE}(\mathbf{y}, \mathbf{y}') \qquad (13)$$

This objective is optimized iteratively through gradient descent to obtain the optimal weight vector $\omega^*$ that gives the insights into the positive/negative impact of each semantic attribute on classification.

Finally, we compute $l_n^{sem}$ as follows:

$$\tilde{\mathbf{S}}^q = \mathbf{S}^q + \omega^*$$
$$l_n^{sem} = \frac{1}{|\tilde{\mathbf{S}}_n^q|} \sum_{\mathbf{s} \in \tilde{\mathbf{S}}_n^q} \mathbf{s} \qquad (14)$$

where $\tilde{\mathbf{S}}_n^q$ is a set of similarity logits in $\tilde{\mathbf{S}}^q$ associated with semantic attributes of class $n$.

## 4 EXPERIMENTS

To test the performance of FewVS, we conducted experiments on four public datasets including *mini*ImageNet, *tiered*ImageNet, CIFAR-FS, and FC100.

### 4.1 Datasets

*mini*ImageNet [42] and *tiered*ImageNet [37] are both subsets of ImageNet [12], while CIFAR-FS [3] and FC100 [31] are derived from CIFAR100 [22]. **miniImageNet** contains 100 classes, with 600 images in each class, and is split into 64 classes for training, 16 classes for validation, and 20 classes for testing. **tieredImageNet** encompasses 351 training classes, 97 validating classes, and 160

**Table 1: Comparison with previous methods on *mini*ImageNet and *tiered*ImageNet. The best performance is highlighted in bold.**

| | Method | Backbone | *mini*ImageNet | | *tiered*ImageNet | |
|---|---|---|---|---|---|---|
| | | | 5-way 1-shot | 5-way 5-shot | 5-way 1-shot | 5-way 5-shot |
| Vision-based | MatchNet [42] | Res12 | 65.64±0.20 | 78.72±0.15 | 68.50±0.92 | 80.60±0.71 |
| | ProtoNet [39] | Res12 | 62.39±0.21 | 80.53±0.14 | 68.23±0.23 | 84.03±0.16 |
| | MAML [16] | Res12 | 49.24±0.21 | 58.05±0.10 | 67.92±0.17 | 72.41±0.20 |
| | CTM [25] | Res12 | 64.12±0.82 | 80.51±0.13 | 68.41±0.39 | 84.28±1.73 |
| | RFS [41] | Res12 | 62.02±0.63 | 79.64±0.44 | 69.74±0.72 | 84.41±0.55 |
| | FEAT [47] | Res12 | 66.78±0.20 | 82.05±0.14 | 70.80±0.23 | 84.79±0.16 |
| | Meta-Baseline [9] | Res12 | 63.17±0.23 | 79.26±0.17 | 68.62±0.27 | 83.29±0.18 |
| | SUN [14] | ViT-S | 67.80±0.45 | 83.25±0.30 | 72.99±0.50 | 86.74±0.33 |
| | FGFL [10] | Res12 | 69.14±0.80 | 86.01±0.62 | 73.21±0.88 | 87.21±0.61 |
| | Meta-AdaM [40] | Res12 | 59.89±0.49 | 77.92±0.43 | 65.31±0.48 | 85.24±0.35 |
| | FewTURE [19] | Swin-T | 72.40±0.78 | 86.38±0.49 | 76.32±0.87 | 89.96±0.55 |
| | SMKD [18] | ViT-S | 74.28±0.18 | 88.82±0.09 | 78.83±0.20 | 91.02±0.12 |
| Semantics-based | KTN [34] | Conv-128 | 64.42±0.72 | 74.16±0.56 | 74.16±0.56 | - |
| | AM3 [44] | Res12 | 65.30±0.49 | 78.10±0.36 | 69.08±0.47 | 82.58±0.31 |
| | TRAML [24] | Res12 | 67.10±0.52 | 79.54±0.60 | - | - |
| | AM3-BERT [46] | Res12 | 68.42±0.51 | 81.29±0.59 | 77.03±0.85 | 87.20±0.70 |
| | SVAE-Proto [45] | Res12 | 74.84±0.23 | 83.28±0.40 | 76.98±0.65 | 85.77±0.50 |
| | SP-CLIP [8] | Visformer-T | 72.31±0.40 | 83.42±0.30 | 78.03±0.46 | 88.55±0.32 |
| | BMI [26] | Res12 | 77.01±0.34 | 84.85±0.27 | 78.37±0.44 | 86.30±0.32 |
| | SemFew-Res [50] | Res12 | 77.63±0.63 | 83.04±0.48 | 78.96±0.80 | 85.88±0.62 |
| | SemFew-Trans [50] | Swin-T | 78.94±0.66 | 86.49±0.50 | 82.37±0.77 | 89.89±0.52 |
| Ours | Baseline-Res | Res12 | 63.31±0.44 | 80.66±0.32 | 69.50±0.50 | 84.23±0.34 |
| | FewVS-Res | Res12 | 82.01±0.29 | 85.29±0.25 | 84.17±0.36 | 87.51±0.30 |
| | Baseline-Trans | ViT-S | 73.43±0.43 | 87.60±0.27 | 77.03±0.52 | 89.84±0.32 |
| | FewVS-Trans | ViT-S | **86.80±0.28** | **90.32±0.22** | **87.87±0.36** | **92.27±0.26** |

testing classes, with 779165 images in total. **CIFAR-FS** contains 100 classes, split into 64 classes for training, 16 classes for validation, and 20 classes for testing. **FC100** introduces a distinctive class partitioning approach, resulting in 60 classes for training, 20 classes for validation, and 20 classes for testing. For both CIFAR-FS and FC100, each class has 600 images with a smaller resolution (32×32) compared with ImageNet.

## 4.2 Implementation Details

**Architecture.** In all experimental configurations, we utilized ResNet-12 (Res12) and ViT-Small (ViT-S) as the few-shot encoders. Specifically, we implemented and pretrained the Res12 encoder using the same settings as described in FEAT [47]. For the ViT-S encoder, we employed the training strategy reported in SMKD [18]. We established two baseline methods using these vanilla few-shot encoders, namely Baseline-Res and Baseline-Trans, in which we performed prototype-based classification following [39]. For the Res12 encoder, visual features were derived by averaging the outputs from the final residual block. For the ViT-S encoder, visual features were obtained by averaging the hidden states from the last transformer block. Additionally, we utilized both the vision and text encoders from Res50x4 CLIP, and employed a single linear layer to construct the linear projection layer $h(\cdot)$.

**Training details.** During the training phase of FewVS, we kept the pretrained few-shot encoder and CLIP's vision encoder frozen and focused only on optimizing the linear projection layer $h(\cdot)$. For FewVS based on Res12, we adhered to traditional FSIC methods [9, 31, 39, 47, 53] by resizing the input images to $84 \times 84$. We also adjusted the stride of the last downsampling layer in CLIP's vision encoder to 1 to accommodate smaller input sizes. For FewVS based on ViT-S, we resized the input image to $320 \times 320$ for *mini*Imagenet and *tierd*Imagene, and $224 \times 224$ for CIFAR-FS and FC100, maintaining the consistency with SMKD [18]. We set the batchsize $B = 128$ and trained the linear projection layer for 20 epochs, using the AdamW optimizer [28] with a learning rate of 1e-3. To construct the entity embedding set $\mathbf{E}$ used in Eq. (3) and Eq. (4), we encoded the names of 1,000 entities sampled from WordNet [30] via CLIP's text encoder.

**Testing details.** At the testing stage, we mined five fine-grained semantic attributes for each class using GPT-3 (*i.e.*, $M = 5$). We online optimized $\omega$ in Eq. (13) via iterative gradient decent, employing an SGD optimizer with a learning rate of 5e-2. The number of iterations was set to 1 for the 1-shot task, and to 5 for the 5-shot task. Additionally, in FewVS based on Res12, we set $\alpha$ in Eq. (9) to 2 and 3 for the 1-shot and 5-shot tasks, respectively. In FewVS

**Table 2: Comparison with previous methods on CIFAR-FS and FC100. The best performance is highlighted in bold.**

| Method | Backbone | CIFAR-FS | | FC100 | |
|---|---|---|---|---|---|
| | | 5-way 1-shot | 5-way 5-shot | 5-way 1-shot | 5-way 5-shot |
| ProtoNet [39] | Res12 | 72.20±0.70 | 83.50±0.50 | 41.54±0.76 | 57.08±0.76 |
| TADAM [31] | Res12 | - | - | 40.10±0.40 | 56.10±0.40 |
| MetaOptNet [23] | Res12 | 72.80±0.70 | 84.30±0.50 | 47.20±0.60 | 55.50±0.60 |
| MABAS [21] | Res12 | 73.51±0.92 | 85.65±0.65 | 42.31±0.75 | 58.16±0.78 |
| RFS [41] | Res12 | 71.50±0.80 | 86.00±0.50 | 42.60±0.70 | 59.10±0.60 |
| Mata-AdaM [40] | Res12 | - | - | 41.12±0.49 | 56.14±0.49 |
| SUN [14] | ViT-S | 78.37±0.46 | 88.84±0.32 | - | - |
| FewTURE [19] | Swin-T | 77.76±0.81 | 88.90±0.59 | 47.68±0.78 | 63.81±0.75 |
| SMKD [18] | ViT-S | 80.08±0.18 | 90.63±0.13 | 50.38±0.16 | 68.37±0.16 |
| SP-CLIP [8] | Visformer-T | 82.18±0.40 | 88.24±0.32 | 48.53±0.38 | 61.55±0.41 |
| SemFew-Res [50] | Res12 | 83.65±0.70 | 87.66±0.60 | 54.36±0.71 | 62.79±0.74 |
| SemFew-Trans [50] | Swin-T | 84.34±0.67 | 89.11±0.54 | 54.27±0.77 | 65.02±0.72 |
| Baseline-Res | Res12 | 71.76±0.49 | 86.27±0.34 | 42.68±0.41 | 59.52±0.42 |
| FewVS-Res | Res12 | 84.40±0.35 | 88.14±0.31 | 58.86±0.40 | 65.27±0.38 |
| Baseline-Trans | ViT-S | 79.47±0.45 | 90.22±0.33 | 48.93±0.43 | 67.29±0.42 |
| FewVS-Trans | ViT-S | **85.63±0.37** | **90.73±0.33** | **61.01±0.40** | **70.37±0.39** |

based on ViT-S, we set $\alpha$ to 2 and 10 for the 1-shot and 5-shot tasks, respectively.

**Evaluation.** To ensure the stability of our evaluation results, we tested on 2,000 randomly sampled episodes from each benchmark test set, and reported the average performance with a 95% confidence interval. In each episode, we randomly selected 15 query features per class to assess the performance in both the 5-way 1-shot and 5-way 5-shot tasks.

## 4.3 Results

The classification results are reported in Table 1 and Table 2. From Table 1 and Table 2, compared with the two baselines, the performance of FewVS is remarkably better. In particular, in the 1-shot task, FewVS achieves accuracy improvement by 13.3% ∼ 18.7%, 10.8% ∼ 14.6%, 6.1% ∼ 12.6%, and 12.1% ∼ 16.2% on *mini*ImageNet, *tiered*ImageNet, CIFAR-FS, and FC100, respectively. This is because the semantic information derived from class names effectively assists in constructing knowledge of novel classes in the extreme absence of support samples.

Compared with other semantics-based competitors, FewVS-Trans surpasses the existing SoTA method (*i.e.*, SemFew [50]) by 0.5% ∼ 7.8% on the four datasets. This is because FewVS enforces unbiased alignment, mines fine-grained semantic attributes, and adaptively integrates these semantic attributes with visual features via an online optimization strategy. In summary, FewVS-Trans achieves new SoTA performance compared with previous FSIC methods across both 1-shot and 5-shot tasks.

## 4.4 Ablation Studies

In this section, our analysis is mainly based on the large challenging dataset, *i.e.*, *tiered*ImageNet, and all the experiments were performed using Res12 as the few-shot encoder. **More ablation studies, such as experiments w.r.t. hyperparameters and visualization analysis, are provided in the supplementary material.**

**Table 3: Evaluation on the optimal transport-based alignment.**

| Method | 1-shot | 5-shot |
|---|---|---|
| w/o alignment | 69.50±0.50 | 84.23±0.34 |
| with $\mathcal{L}_{L2}$ | 80.01±0.41 | 86.91±0.31 |
| FewVS | 84.17±0.36 | 87.51±0.30 |

### 4.4.1 Impact of the optimal transport-based alignment.
In FewVS, we proposed the optimal transport-based alignment, which maximized the consistency between assignment predictions computed from the output features of the few-shot encoder and those of CLIP's vision encoder. To study the impact of this alignment, we tested two variants. In the first variant (**w/o alignment**), we removed the alignment module and degenerated FewVS into a baseline. In the second variant (**with $\mathcal{L}_{L2}$**), we defined the alignment objective as the minimization of the L2 distance between the features from the two encoders. As shown in Table 3, compared with FewVS, the performance of these two variants drops significantly, verifying the effectiveness of the optimal transport-based alignment.

### 4.4.2 Impact of the entity-based cluster centers in assignment prediction.
Different from previous optimal transport-based approaches [1, 2, 5, 33], we constructed the cluster centers (*i.e.*, **E** in Eq. (3) and Eq. (4)) for feature assignment using entity embeddings. These embeddings were encoded from the names of entities in WordNet through the text encoder of CLIP. To evaluate the efficacy of these entity-based cluster centers, we tested two variants. In the first variant (**w/o entity embed**), we defined **E** as 3K learnable prototypes following [1, 2, 5]. In the second variant (**with word embed**), we defined **E** as the word embeddings of CLIP's text encoder following [33]. As shown in Table 4, compared with these two

Table 4: Evaluation on the entity-based cluster centers.

| Method | 1-shot | 5-shot |
|---|---|---|
| w/o entity embed. | 68.23±0.47 | 82.28±0.35 |
| with word embed. | 79.19±0.39 | 87.36±0.31 |
| FewVS | 84.17±0.36 | 87.51±0.30 |

Table 5: Evaluation on mining fine-grained semantic attributes.

| Method | 1-shot | 5-shot |
|---|---|---|
| w/o mining | 79.13±0.44 | 85.36±0.33 |
| FewVS | 84.17±0.36 | 87.51±0.30 |

Table 6: Evaluation on the online vision-semantics integrator.

| Method | 1-shot | 5-shot |
|---|---|---|
| w/o optimization | 83.03±0.39 | 85.97±0.34 |
| FewVS | 84.17±0.36 | 87.51±0.30 |

variants, the entity-based cluster centers achieve substantial performance improvement, which is possibly because the cluster centers constructed from entities are better to represent image contents, thus producing more accurate feature assignments.

*4.4.3 **Impact of mining fine-grained semantic attributes***. In FewVS, we mined fine-grained semantic attributes from GPT-3 to enrich the semantic information of class names. To demonstrate its effectiveness, we tested a variant (**w/o mining**), in which we removed the semantic attribute mining process. In this variant, the semantic prototype $\hat{p}(n)$, defined in Eq. (11), only relies on the semantic feature encoded from the class name $n$. Therefore, there is no need to re-weight the semantic attributes, and the classification is straightforwardly conducted using Eq. (11). As shown in Table 5, the performance of FewVS notably deteriorates in the absence of fine-grained semantic attributes, indicating that mining semantic attributes from large language models does enrich semantic information on class names.

*4.4.4 **Impact of the online vision-semantics integrator***. In FewVS, we designed an online vision-semantics integrator that adaptively integrated vision and semantics by re-weighting the classification contribution of each semantic attribute. This was accomplished by iteratively optimizing $\omega$ in Eq. (13). To investigate the rationality of this module, we tested a variant (**w/o optimization**), in which the number of optimization iterations was set to 0. From Table 6, FewVS involving online optimization performs better than its variant. This demonstrates the rationality of the online vision-semantics integrator, which can identify the positive/negative impact of each semantic attribute on classification.

Table 7: Comparison results on *tiered*ImageNet.

| Method | Backbone | # Params | 1-shot | 5-shot |
|---|---|---|---|---|
| Tip-Adapter | Res50x4 | 87M | 82.24 ±0.35 | 82.95±0.34 |
| Baseline | Res12 | 12M | 69.50±0.50 | 84.23±0.34 |
| FewVS | Res12 | 12M | **84.17±0.36** | **87.51±0.30** |

*4.4.5 **Compared with CLIP-based methods***. We compared our FewVS with a CLIP-based approach, *i.e.*, Tip-Adapter [52]. Tip-Adapter adapts CLIP to few-shot scenarios using a training-free key-value cache model [20, 32]. The original few-shot setting in Tip-Adapter differs from general FSIC. Therefore, we modified Tip-Adapter for FSIC by restricting the cache model to only store images from an $N$-way $K$-shot episode as the key-value database during the testing phase.

As reported in Table 7, FewVS outperforms Tip-Adapter by a large margin on *tiered*ImageNet and has fewer parameters. This is because compared with the heavy CLIP vision encoder (namely Res50x4), FewVS utilizes a lightweight few-shot encoder. This encoder can be easily fine-tuned or trained from scratch using methodologies specifically designed for FSIC, facilitating the encoding of task-specific visual information.

## 5 CONCLUSION

In this paper, we proposed a framework, called FewVS, to integrate visual and semantic information for improving FSIC. In FewVS, we indirectly achieved unbiased vision-semantics alignment by introducing CLIP and enforcing vision-vision alignment as a proxy task, with an optimal transport-based objective. Furthermore, we mined fine-grained semantic attributes from large language models, and designed an online optimization module to adaptively integrate these semantic attributes with information extracted from images. Experiments showed that FewVS effectively improved baseline methods and outperformed the state-of-the-art methods on four challenging datasets.

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
