# OpenReview forum: "FewVS: A Vision-Semantics Integration Framework for Few-Shot Image Classification"
_acmmm.org/ACMMM/2024/Conference — MM2024 Poster_

### Official Review · Reviewer_mPNL · 2024-05-18

**Rating:** 4
**Confidence:** 3

**Summary:**

In this paper, the authors propose a few-shot vision semantics integration framework (FewVS). FewVS addresses data scarcity and insufficient information fusion issues by aligning a pre-trained few-shot encoder with CLIP’s visual encoder on the same image, thereby indirectly aligning the few-shot encoder with CLIP’s text encoder.

**Strengths:**

1. The proposed method addresses the challenge of few-shot learning by enriching the feature space with language-derived context, which is often lacking in purely visual methods.
2. The writing in this paper is clear and effectively communicates the proposed framework and experimental results. The authors also provide clear explanations of the methodology.
3. The proposed method is theoretically reasonable using feature alignment based on optimal transmission. This method also ensures accurate feature alignment between the few-shot encoder and the CLIP visual encoder, thereby enhancing the effectiveness of FewVS.

**Limitations:**

1. The FewVS framework relies on large pre-trained models (such as CLIP and GPT-3) for visual and semantic integration. This dependency may result in high computational costs and resource requirements.
2. Although the paper introduces the FewVS framework, its approach mainly involves combining existing technologies rather than introducing genuinely novel or innovative ideas or techniques. For example, while advanced models like CLIP are used, the integration method does not introduce significant innovation and lacks breakthrough improvements compared to some previous work.
3. The FewVS framework relies on the CLIP model for visual and semantic integration, but does not address or remove noise in the CLIP model. As large language models (such as CLIP) may contain noise, this could affect the accuracy of final feature alignment and the quality of integration.
4. Some references have inconsistent and non-standard citation formats. For instance, the conference names in reference [36] are capitalized while they are lowercase in reference [31]. Reference [36] uses conference abbreviations, whereas the full names are used in reference [42]. Some references lack page numbers, suggesting careful revision is needed.

**Suitability:**

2

---

### Official Review · Reviewer_GEEh · 2024-05-23

**Rating:** 3
**Confidence:** 2

**Summary:**

This paper proposes a proxy task of vision-vision alignment to improve the effectiveness of CLIP for few-shot classification, where the proxy task is to align the visual features from the CLIP and few-shot encoder. In this way, the visual features can achieve a better alignment with the text encoder of CLIP. Furthermore, an online vision-semantics integrator is introduced to provide a weight for feature fusion.

**Strengths:**

1. The motivation of the paper is clear and convincing;
2. The idea of vision-vision alignment is a neat way to better use CLIP;
3. The proposed method is effective with a significant improvement.

**Limitations:**

1. It seems that the core idea of this paper is a method similar to CLIP distillation, so the contribution appears to be limited;
2. I wonder if is there a comparison of directly inputting the CLIP's visual features into the online vision-semantics integrator;
3. Both vision-vision alignment and fine-grained semantic attributes mining are convincing ideas, but these techniques may have already been introduced in previous works or other tasks, lacking specific improvements and modifications for few-shot image classification accordingly;
4. The ablation studies seems to have been conducted only on the ResNet backbone.

**Suitability:**

2

---

### Official Review · Reviewer_q4vz · 2024-05-24

**Rating:** 3
**Confidence:** 2

**Summary:**

This paper tackles the few-shot image classification problem by aligning with CLIP from two perspectives. First, the method utilizes vision-vision alignment with a training objective formulated using optimal transport-based assignment prediction as a proxy task to avoid biased vision-semantics alignment. Further, the method proposes to use fine-grained semantic attributes instead of the single class name for classification. The proposed method achieve SOTA performance on four datasets.

**Strengths:**

1. The paper is easy to follow.
2. The proposed method achieves SOTA performance on various benchmarks.

**Limitations:**

1. This work aims to align with CLIP in an unbiased way. Why not directly use the CLIP visual encoder to extract the feature as CLIP has shown impressive zero-shot image classification ability. Finetuning CLIP with some adapters or other techniques could possibly be another solution. In Table 7, the proposed method outperforms than a CLIP-based approach Tip-Adapter. Unfortunately, there is no detailed explanation.
2. This paper performs the vision-vision alignment at the output of the encoders. Perhaps aligning intermediate feature maps can provide more useful details. Typically, in the few shot setting, aligning with more samples can significantly increase the performance as shown in Table 3. Thus, exploring the intermediate feature map alignment is reasonable. Moreover, I think aligning with CLIP in the pretraining phase can further improve the performance.
3. The pretrained few-shot encoder is frozen during the training which is not reasonable as only one projection layer may not be sufficient to align with CLIP. Perhaps more tunable parameters can better align the model with CLIP.
4. More detailed captions under Figure 1 and Figure 3 can help readers better understand the total pipeline.
5. There are some typos in the manuscript, for example \hat{x}^q \in R^{\hat{d}} in Line 403. Please check them carefully.

**Suitability:**

3

---

### Official Review · Reviewer_bWYb · 2024-05-24

**Rating:** 5
**Confidence:** 3

**Summary:**

This paper aims to align the few-shot feature encoder with the CLIP vision encoder to indirectly align with CLIP's text encoder. The authors propose that this approach addresses the issue of biased vision-semantics alignment.

**Strengths:**

1. The paper is well-written and easy to understand.

2. The novel idea of leveraging CLIP to bridge the gap between vision and semantics is simple, interesting, and intuitive.

3. The experiments are extensive, and the proposed method achieves promising performance in few-shot image classification across four datasets.

**Limitations:**

1) From the experiment, I observed that the Swin-T backbone performs well across the four datasets. Have the authors tried using the proposed method based on this backbone?

2) In the table, with L2, the 1-shot performance drops significantly, but the 5-shot performance does not drop as much. It would be interesting to provide more analysis on this.

**Suitability:**

3

---

### Meta-Review · Program_Chairs · 2024-07-01

**Recommendation:** Accept (Poster)
**Confidence:** 4

**Metareview:**

This paper introduces CLIP, a bridge between vision and semantics.

The reviewers highlight a number of relevant issues, including limited novelty and clarity. Nevertheless, based on the rebuttal, all reviewers converged in borderline accept (with two of them moving from reject to accept). We encourage the authors to take into account all the feedback for the camera-ready version of the article.